# A Novel High-Sensitivity MEMS Pressure Sensor for Rock Mass Stress Sensing

**DOI:** 10.3390/s22197593

**Published:** 2022-10-07

**Authors:** Honghui Wang, Dingkang Zou, Peng Peng, Guangle Yao, Jizhou Ren

**Affiliations:** 1State Key Laboratory of Geohazard Prevention and Geoenvironment Protection, Chengdu University of Technology, Chengdu 610059, China; 2College of Computer Science and Cyber Security, Chengdu University of Technology, Chengdu 610059, China

**Keywords:** rock stress sensing, piezoresistive pressure sensor, MEMS, simulation

## Abstract

This paper proposes a novel high-sensitivity micro-electromechanical system (MEMS) piezoresistive pressure sensor that can be used for rock mass stress monitoring. The entire sensor consists of a cross, dual-cavity, and all-silicon bulk-type (CCSB) structure. Firstly, the theoretical analysis is carried out, and the relationship between the structural parameters of the sensor and the stress is analyzed by finite element simulation and curve-fitting prediction, and then the optimal structural parameters are also analyzed. The simulation results indicate that the sensor with the CCSB structure proposed in this article obtained a high sensitivity of 87.74 μV/V/MPA and a low nonlinearity error of 0.28% full-scale span (FSS) within the pressure range of 0–200 MPa. Compared with All-Si Bulk, grooved All-Si Bulk, Si-Glass Bulk, silicon diaphragm, resistance strain gauge, and Fiber Bragg grating structure pressure sensors, the designed sensor has a significant improvement in sensitivity and nonlinearity error. It can be used as a new sensor for rock disaster (such as collapse) monitoring and forecasting.

## 1. Introduction

Traditional measurement techniques, such as multiple-point extensometers, surface displacement sensing, and global positioning systems (GPS), are useful for monitoring the surface deformation of geological hazards, such as landslides [1]. For the failure and instability of the rock slope, the stress concentration in the rock mass exceeds the failure stress of the rock mass itself, which causes the rock mass to rupture and then collapse and fall. Generally speaking, the macroscopic cracks and deformation failures of rock masses tend to remain after the initiation, coalescence, and expansion of microcracks inside the rock mass. The sudden and “unheralded” occurrence of rock slope and collapse makes the traditional monitoring methods mentioned above unable to provide early warning of the occurrence of these disasters. The development of micro-cracks in the rock mass is inevitably related to the change in rock mass stress [2]. Therefore, the rock slope instability can be warned by monitoring the rock mass stress. It is very interesting to develop a sensor suitable for sensing the stress of rock mass.

Rock slope stress monitoring requires sensors with high sensitivity and a large measuring range. Examples of current rock stress monitoring cases include the following: Su et al. have developed a sensor for quantitative measurement of rock mass stress and strain, which can measure the pressure and deformation of the measured point of the rock mass, but the maximum range of the sensor is only 10 Kg [3]. Zhao et al. proposed a borehole deformation sensor based on fiber grating [4], which uses four rings to convert borehole deformation into fiber strain, which is used for long-term monitoring of coal mine rock mass stress with a measurement accuracy of about 300~500 μm. Luna’s fiber optic os9100 sensors are low-profile Fiber Bragg grating (FBG)-based discrete static and dynamic pressure sensors, with a pressure range of −3~13 KPa, a resolution of 2 Pa, and a non-linearity of 1.0%. It can be deployed in large numbers and installed over long distances.

Compared with traditional pressure sensors composed of metal strain gauges, MEMS pressure sensors are made based on semiconductor materials and processing technology, such as silicon, which makes these sensors have the advantages of high sensitivity, low cost, miniaturization, and easy integration. According to the working principle of pressure sensors, MEMS pressure sensors can be divided into piezoresistive, piezoelectric, capacitive, resonant, and other types. 

Material selection: MEMS pressure sensors can be divided into silicon, carbon nanotube (CNT), graphene, and so on. Silicon is the preferred material for MEMS piezoresistive pressure sensors with high sensitivity, repeatability, and high mechanical properties, and the current silicon manufacturing process is mature and low in cost. However, the silicon-doped piezoresistance is isolated by the P–N junction and is easily affected by temperature. At higher temperatures, there will be a greater leakage current, which will cause the sensor performance to decrease [5].

Carbon nanotubes (CNTs) have unique advantages, i.e., excellent mechanical properties, high electrical conductivity, and thermal stability. As a piezoresistive film, a large-area flexible strain sensor with high sensitivity and low manufacturing cost can be realized. It allows strain measurement for both integral measurement on a certain surface and local measurement at a certain position depending on the sensor geometry [6].

Graphene has excellent electrical conductivity, super flexibility, and stretchability of up to 20%, Zhu et al. integrated graphene resistors on silicon nitride (SiNx) membranes and developed a graphene-based piezoresistive pressure sensor with an external pressure of 500 mbar and a sensitivity of 8.5 mV/bar [7]. Smith et al. proposed a piezoresistive graphene sensor with a maximum pressure of 100 KPa, which is more sensitive than silicon and CNT-based sensors [8]. 

Among them, the silicon piezoresistive pressure sensor is the most commercialized and most widely used MEMS pressure sensor. Taking into account factors such as low cost and ease of manufacturing, this study uses silicon as the material for the piezoresistive and core structure of the pressure sensor. The higher the natural frequency of the sensor, the higher the sensitivity of the sensor will be. The bearing capacity can be improved by increasing the structure.

Structure design: The diaphragm-type structure is welcomed by industry engineers and researchers. Huang et al. proposed a peninsula-structured diaphragm-type piezoresistive sensor [9]. In comparison to a flat diaphragm, the proposed sensor design could achieve a sensitivity increase by 11.4% in the pressure range of 0–5 kPa. The cross beam-membrane (CBM) structure piezoresistive pressure sensor used for micro pressure measurement has the advantages of high sensitivity, high linearity, and high accuracy. The experiment shows that the sensitivity of the CBM sensor is 7.081 mv/KPa, and the nonlinear error is 0.09% FSS under the pressure load of 0–5 KPa [10]. In order to promote the sensing range of the diaphragm-type sensor, Niu et al. designed a square diaphragm-type piezoresistive pressure sensor suitable for high pressure and high-temperature environments, with a pressure range of 0–250 MPa and a sensitivity of 1.1126 mV/MPa [11]. Nag et al. designed a novel structure by introducing the local stiffness in the diaphragm membrane for low-pressure measurement, the rod beams at the diaphragm with combination of graphene piezoresistors, improve overall performance of the pressure sensor in terms of sensitivity [12]. Li et al. proposed a novel structural piezoresistive pressure sensor with a four-beams-structured membrane, which has achieved a sensitivity of 25.48 mV/kPa and a nonlinearity error of 0.75% FSS, but the pressure measurement range is less than 5 kPa [13]. Overall, a large number of researches on diaphragm sensors are mainly used for the measurement of low-pressure or micro-pressure ranges [14,15,16,17], but it is obviously unsatisfactory for some high-pressure application scenarios, such as rock fracture stress, in situ stress, oil and gas exploitation, etc. The range of rock stress is related to the composition of the rock. Generally, the compressive strength of rock mass varies from 10–100 MPa, while for the scenario of oil and gas exploitation, the pressure on the bit is more than 100 MPa [18].

Since the movable diaphragm structure has the congenital defects of vulnerability and overload resistance, Heinickel et al. proposed a silicon–glass bulk-type piezoresistive pressure sensor with a pressure range of up to 500 MPa and strong overload resistance with a sensitivity of 22.24 μ/V/MPa [19]. Kevin Chau’s team proposed a bulk-type all-silicon pressure sensor, which can achieve a displacement of 200 MPa and a large overload capacity, with a sensitivity of 79 μV/V/MPa [20,21,22]. Based on the All-Si Bulk structure, Lin et al. proposed an All-Si Bulk with Trench type, that is placing stress filtering trenches near the selected varistor pair to convert the local stress field from biaxial to uniaxial to eliminate the symmetry of stress [22]. Additionally, the piezoresistance subjected to uniaxial stress is very sensitive to the applied stress.

In summary, the diaphragm-type structure has more advantages in sensitivity than the All-Si Bulk structure. The All-Si Bulk structure has no movable diaphragm and has the potential to withstand greater pressure than the diaphragm-type structure. However, the slope rock mass stress varies according to the nature of the rock mass, and it is often above several MPa to tens of MPa [18]. In order to shorten the warning time, it is necessary to develop a high-sensitivity rock mass stress sensor. The sensitivity and other performances of the existing All-Si Bulk sensors have encountered some challenges in the scenario mentioned above. Therefore, this study mainly focused on researching the optimal All-Si Bulk structure, and designed a cross, dual-cavity, and all-silicon bulk-type (CCSB) sensor structure, which can not only meet the demands of large ranges, but also meet the needs of higher sensitivity for the rock mass stress monitoring.

This research involves designing a pressure sensor with a cross, dual-cavity, and all-silicon bulk-type (CCSB) structure, which is suitable for a high-pressure range for rock mass stress monitoring. First, the piezoresistive coefficient of the sensor in various directions is theoretically analyzed, and the direction of the piezoresistive position is determined. Then, the relationship between the structural parameters and stress of the sensor is deduced through finite element simulation and MATLAB, and the optimal design of the sensor size is obtained according to the analysis results. Lastly, the performance of the sensor with other structures is compared through finite element simulation.

## 2. Principle of CCSB

### 2.1. The CCSB Structural Features

The all-silicon bulk-type pressure sensor is based on the principle of piezoresistive effect. The biggest difference to the traditional diaphragm type is that we abandoned the design of the diaphragm type that directly bears external pressure. The all-silicon structure uses a “cap”-like design to directly withstand external pressure (Figure 1). The advantage of this design is that the “cap” can withstand greater pressure and the sensor also has a stronger overload capacity. By making a cavity at the top of the piezoresistive placement area inside the “cap”, the piezoresistance is only subjected to plane stress in its plane, and there is no vertical stress. Zeng et al. [20] have analyzed the three stress conditions caused by three different structures, such as: triaxial compression (without cavity), biaxial compression (with cavity), and uniaxial compression (piezoresistive placed on the bridge with cavity). The beam-membrane structure containing the cross is a biaxial compression structure. The stress in the piezoresistive placement area can be increased by realizing a cavity in the base, and compared with the flat membrane structure, placing the piezoresistance on the cross structure can increase the stress it receives. The “cap” of the sensor is subjected to hydrostatic pressure in five directions outside, and the base is also subjected to hydrostatic pressure in all directions in addition to the bottom fixing.

### 2.2. Piezoresistive Effect of CCSB

When single crystal silicon is subjected to stress, the resistivity of silicon is anisotropic. It is related to the piezoresistive coefficient (*π**_ij_*) and stress tensor (*σ**_ij_*). For silicon with a cubic crystal structure, the piezoresistive coefficient has only three non-zero components [23]. Equation (1) is the piezoresistive coefficient and stress tensor in the crystal coordinate system.
(1)$$\frac{1}{\rho_{0}}\left[\begin{array}{l} \Delta \rho_{11}\\ \Delta \rho_{22}\\ \Delta \rho_{33}\\ \Delta \rho_{23}\\ \Delta \rho_{13}\\ \Delta \rho_{12} \end{array}\right]=\left[\begin{array}{cccccc} \pi_{11}^{} & \pi_{12}^{} & \pi_{12}^{} & 0 & 0 & 0\\ \pi_{12}^{} & \pi_{11}^{} & \pi_{12}^{} & 0 & 0 & 0\\ \pi_{12}^{} & \pi_{12}^{} & \pi_{11}^{} & 0 & 0 & 0\\ 0 & 0 & 0 & \pi_{44}^{} & 0 & 0\\ 0 & 0 & 0 & 0 & \pi_{44}^{} & 0\\ 0 & 0 & 0 & 0 & 0 & \pi_{44}^{} \end{array}\right]\cdot \left[\begin{array}{c} \sigma_{11}^{}\\ \sigma_{22}^{}\\ \sigma_{33}^{}\\ \sigma_{23}^{}\\ \sigma_{13}^{}\\ \sigma_{12}^{} \end{array}\right]$$
where *ρ*_0_ is the initial piezoresistance, Δ*ρ* is the change in resistivity, *π* is the piezoresistance coefficient and *σ* is the stress tensor.

As shown in Figure 1, the surface of the sensor varistor is located on the (100) plane, the piezoresistive current direction is along the [110] crystal direction, and the piezoresistive vertical current direction is along the $[1\bar{1}0]$ crystal direction. The size of the piezoresistive coefficient is related to factors such as crystal orientation, doping type, and temperature, so the piezoresistive coefficient of piezoresistance on the (100) crystal plane changes with the direction. Because of the cavity, only transverse stress *σ*_11_ and longitudinal stress *σ*_22_ exist on the device surface. Then, Equation (1) can be derived:(2)$$\frac{1}{\rho_{0}}\left[\begin{array}{l} \Delta \rho_{11}\\ \Delta \rho_{22}\\ \Delta \rho_{33}\\ \Delta \rho_{23}\\ \Delta \rho_{13}\\ \Delta \rho_{12} \end{array}\right]=\left[\begin{array}{cccccc} \pi_{11}^{\prime } & \pi_{12}^{\prime } & \pi_{13}^{\prime } & \pi_{14}^{\prime } & \pi_{15}^{\prime } & \pi_{16}^{\prime }\\ \pi_{21}^{\prime } & \pi_{22}^{\prime } & \pi_{23}^{\prime } & \pi_{24}^{\prime } & \pi_{25}^{\prime } & \pi_{26}^{\prime }\\ \pi_{31}^{\prime } & \pi_{32}^{\prime } & \pi_{33}^{\prime } & \pi_{34}^{\prime } & \pi_{35}^{\prime } & \pi_{36}^{\prime }\\ \pi_{41}^{\prime } & \pi_{42}^{\prime } & \pi_{43}^{\prime } & \pi_{44}^{\prime } & \pi_{45}^{\prime } & \pi_{46}^{\prime }\\ \pi_{51}^{\prime } & \pi_{52}^{\prime } & \pi_{53}^{\prime } & \pi_{54}^{\prime } & \pi_{55}^{\prime } & \pi_{56}^{\prime }\\ \pi_{61}^{\prime } & \pi_{62}^{\prime } & \pi_{63}^{\prime } & \pi_{64}^{\prime } & \pi_{65}^{\prime } & \pi_{66}^{\prime } \end{array}\right]\cdot \left[\begin{array}{c} \sigma_{11}^{\prime }\\ \sigma_{22}^{\prime }\\ 0\\ 0\\ 0\\ \sigma_{12}^{\prime } \end{array}\right]$$
where *π′_ij_* is the piezoresistive coefficient in any coordinate system. *σ′*_11_ and *σ′*_22_ are the plane normal stress in any coordinate system and *σ′*_12_ is the plane shear stress in any coordinate system. The direction of the long axis of the piezoresistance is the direction of the current. Additionally, without considering the current situation in other directions, under the action of stress, the relative change value of piezoresistive resistivity can be simplified as:(3)$$\frac{\Delta \rho_{11}}{\rho_{0}}=\pi_{11}^{\prime }\sigma_{11}^{\prime }+\pi_{12}^{\prime }\sigma_{22}^{\prime }+\pi_{16}^{\prime }\sigma_{12}^{\prime }$$

On the (100) plane, *π′*_11_, *π′*_12_, *π′*_16_ under any piezoresistive direction can be calculated as [24]:(4)$$\begin{array}{l} \pi_{11}^{\prime }=\pi_{11}-2(\pi_{11}-\pi_{12}-\pi_{44})(l_{1}^{2}m_{1}^{2}+l_{1}^{2}n_{1}^{2}+n_{1}^{2}m_{1}^{2})\\ \pi_{12}^{\prime }=\pi_{12}+(\pi_{11}-\pi_{12}-\pi_{44})(l_{1}^{2}l_{2}^{2}+m_{1}^{2}m_{2}^{2}+n_{1}^{2}n_{2}^{2})\\ \pi_{16}^{\prime }=2(\pi_{11}-\pi_{12}-\pi_{44})(l_{1}^{3}l_{2}+m_{1}^{3}m_{2}+n_{1}^{3}n_{2}) \end{array}$$
where *l_i_*, *m_j_* and *n_k_* are the directional cosines of the unit vectors on the *x*, *y*, and *z* axes in any Cartesian coordinate system in the crystal coordinate system, respectively.

For the piezoresistive placed on the (100) plane, the piezoresistance can be placed arbitrarily, that is the crystal coordinate system rotates *θ* degrees around the *Z*-axis.
(5)$$\left[\begin{array}{ccc} l_{1} & m_{1} & n_{1}\\ l_{2} & m_{2} & n_{2}\\ l_{3} & m_{3} & n_{3} \end{array}\right]=\left[\begin{array}{ccc} \cos(\theta ) & \sin(\theta ) & 0\\ -\sin(\theta ) & \cos(\theta ) & 0\\ 0 & 0 & 1 \end{array}\right]$$
i.e.,
(6)$$\begin{array}{l} \pi_{11}^{\prime }=\pi_{11}-2(\pi_{11}-\pi_{12}-\pi_{44})(\cos^{2}\theta \cdot \sin^{2}\theta )\\ \pi_{12}^{\prime }=\pi_{12}+(\pi_{11}-\pi_{12}-\pi_{44})(2\cos^{2}\theta \cdot \sin^{2}\theta )\\ \pi_{16}^{\prime }=2(\pi_{11}-\pi_{12}-\pi_{44})(-\cos^{3}\theta \sin\theta +\sin^{3}\theta \cos\theta ) \end{array}$$

From the piezoresistance coefficient table (Table 1) of single crystal silicon, the relationship between the piezoresistance coefficient and angle of p-type silicon and n-type silicon can be obtained (Figure 2). Since the contact between the metal Al and the p-type doping is an ohmic contact, the contact with the n-type doping is a diode-like rectification characteristic, and the contact with the n+ type is a nonlinear ohmic contact. Therefore, to simplify the lead, the most commonly used piezoresistance is mainly p-type [25].

For the piezoresistive placed on the (100) plane, *σ*_12_ is the shear stress on the (100) plane. Since only biaxial stress exists, *σ*_12_ is 0. As shown in Figure 3, the force analysis of piezoresistance can be obtained from the elastic theory through-plane stress analysis:(7)$$\begin{array}{l} \sigma_{11}^{\prime }=\sigma_{11}\cos^{2}\alpha +\sigma_{22}\sin^{2}\alpha -2\sigma_{12}\sin\alpha \cos\alpha \\ \sigma_{22}^{\prime }=\sigma_{11}\sin^{2}\alpha +\sigma_{22}\cos^{2}\alpha +2\sigma_{12}\sin\alpha \cos\alpha \\ \sigma_{12}^{\prime }=\sigma_{11}\sin\alpha \cos\alpha -\sigma_{22}\sin\alpha \cos\alpha +\sigma_{12}\cos2\alpha \end{array}$$
i.e.,
(8)$$\left[\begin{array}{l} \sigma_{11}^{\prime }\\ \sigma_{22}^{\prime }\\ \sigma_{12}^{\prime } \end{array}\right]=\left[\begin{array}{ccc} \cos^{2}\alpha & \sin^{2}\alpha & -\sin2\alpha \\ \sin^{2}\alpha & \cos^{2}\alpha & \sin2\alpha \\ \frac{\sin2\alpha }{2} & -\frac{\sin2\alpha }{2} & \cos2\alpha \end{array}\right]\cdot \left[\begin{array}{c} \sigma_{11}\\ \sigma_{22}\\ 0 \end{array}\right]$$

Therefore, from Equations (3), (6) and (8), the relative change of resistivity of each piezoresistance under the action of in-plane stress can be obtained.

### 2.3. Sensor Output

Piezoresistive pressure sensors usually use Wheatstone bridge circuits to measure changes in piezoresistance. For four resistors doped on the plane (100), the output of the bridge is (*R*_1_
*= R*_3_, *R*_2_
*= R*_4_):(9)$$\frac{V_{O}}{V_{S}}=\frac{R_{1}R_{3}-R_{2}R_{4}}{\left(R_{1}+R_{2})(R_{3}+R_{4}\right)}$$

Common evaluation criteria for sensors include sensitivity (*S*) and nonlinear error (*NL_i_*), etc.
(10)$$S=\frac{V_{\mathrm{out}}(p_{m})-V_{\mathrm{out}}(p_{1})}{(p_{m}-p_{1})\cdot V_{s}}=\frac{V_{FS}}{(p_{m}-p_{1})\cdot V_{s}}$$
(11)$$NL_{i}=100\%\times \frac{V_{out}(p_{i})-\frac{V_{out}(p_{\max})}{p_{\max}}\cdot p_{i}}{V_{out}(p_{\max})}$$
where *V_out_*(*P_m_*) and *V_out_*(*P*_1_) are the output voltage of the bridge when the external maximum pressure *P_m_* and minimum pressure *P*_1_ are applied, respectively. *V_FS_* and *V_S_* are the full-scale output voltage and the bridge supply voltage, respectively, and *P_i_* is the *i*-th externally applied pressure.
(12)$$\rho =\frac{R\cdot S}{L}$$

Finally, according to Equations (6), (7), (9), (10) and (12), the relationship between the sensor sensitivity and the stress of Equation (13) can be obtained:(13)$$S=\frac{\Delta R/R}{p_{m}-p_{1}}=\frac{\Delta \rho /\rho }{p_{m}-p_{1}}=\frac{\pi_{11}^{\prime }\sigma_{11}^{\prime }+\pi_{12}^{\prime }\sigma_{22}^{\prime }+\pi_{16}^{\prime }\sigma_{12}^{\prime }}{p_{m}-p_{1}}=\frac{\pi_{11}^{}\sigma_{11}^{}+\pi_{12}^{}\sigma_{22}^{}}{p_{m}-p_{1}}$$

It should be pointed out that the in-depth analysis of piezoresistive behavior will be carried out in detail for this structure sensor in the future.

## 3. Sensor Design

### 3.1. The CCSB Stress Analysis

The structure of the all-silicon bulk-type pressure sensor includes an upper cap, a cavity for biaxial compression, and a lower base, which can withstand high pressure and has a strong overload capacity. However, the simple cavity structure of the all-silicon bulk-type results in reduced sensor sensitivity comparing to the traditional diaphragm type, where the pressure is applied directly to the diaphragm. Therefore, this research considers placing the piezoresistance on the diaphragm structure, which contains the bottom cavity structure. 

The finite element software is used to numerically solve the established model, and the stress distribution curves of the piezoresistive regions of the two structures can be obtained (Figure 4).

It is found that the stresses in the core of the bottomless cavity structure in the piezoresistive doping region are concentrated around the upper cavity and the stresses increasing from the inside to the outside, while the stresses near the middle are smaller and vary less. The core with a bottom cavity structure has less fluctuation in stress change from the center of the cavity to the outside, but there is stress concentration around the cavity. The bottom cavity structure distributes the stress concentration around the bottomless cavity structure to the entire plane, increasing the stress inside the cavity bottom. If the piezoresistive strip is doped in an area with uneven or abrupt stress distribution, it will cause uneven stress on the piezoresistance itself and increase the nonlinear error of the sensor. Therefore, the bottom cavity structure can improve the sensitivity of the sensor and reduce its nonlinear error.

The external pressure that the all-silicon bulk-type structure of the cross can withstand is related to the nature of the silicon material. The fracture stress of the silicon material is 7000 MPa, so the ultimate stress of the sensor should be less than the fracture stress, and considering the sensor’s anti-overload design, the sensor pressure is designed as 70% of the maximum pressure. Using finite element simulation (Figure 4c), it can be seen that when *P* is 1200 MPa, the maximum stress of the sensor is close to the fracture limit of silicon material. Thus, in this research, the pressure of the sensor is designed to be 800 MPa, and the maximum strain is 6.29 μm under this pressure.

### 3.2. The CCSB Structure Analysis

#### 3.2.1. Structural Parameter Analysis

As for the relationship between stress and structure, because the variable elements involved are too many and too complex, this paper will fit the stress expression from the perspective of simulation.

Firstly, the COMSOL finite element simulation software was used to predict the influence of each parameter of the cross all-silicon bulk-type structure model on the surface stress of the piezoresistive placement. In this research, the dimensions of the all-silicon bulk-type structure include the length *L*_0_, the width *W*_0_ and height *H*_0_ of the upper cap, the length *L*_1_, the width *W*_1_ and height *H*_1_ of the upper cavity, the length *L*_2_, the width *W*_2_ and height *H*_2_ of the base, the length *L*_3_, the width *W*_3_ and height *H*_3_ of the cross, and the length *L*_4_, the width *W*_4_ and height *H*_4_ of the bottom cavity. The initial design dimensions are shown in Table 2. 

Because it is difficult to directly derive the stress theoretical formula of the crossed all-silicon bulk structure, the combination of finite element calculation and curve fitting is considered to determine the approximate theoretical relationship. Since the stress of the traditional diaphragm structure is the power function relationship of the structural variables [17], the functional form of the all-silicon structure can be approximated as power functions of each structural dimension:(14)$$\sigma_{pzr}=a\cdot L_{0}^{b_{0}}\cdot H_{0}^{b_{1}}\cdot L_{1}^{b_{2}}\cdot H_{1}^{b_{3}}\cdot L_{2}^{b_{4}}\cdot H_{2}^{b_{5}}\cdot W_{3}^{b_{6}}\cdot H_{3}^{b_{7}}\cdot H_{4}^{b_{8}}$$
where *L*_0_ = *W*_0_, *L*_1_ = *W*_1_ = *L*_3_ = *L*_4_, *L*_2_ = *W*_2_.

To determine these constants in Equation (14), each variable should be analyzed while others are held constant. That is, under the condition that other parameters are fixed, the stress results can be simulated by changing the size of a single parameter, and then curve fitting can be performed to determine the coefficient of each parameter variable., i.e.,
(15)$$\left\{ \begin{array}{l} \sigma_{pzr}=a_{L_{0}}\cdot L_{0}^{b_{1}}\\ \sigma_{\mathrm{pzr}}=a_{H_{0}}\cdot H_{0}^{b_{2}}\\ \ldots \ldots \\ \sigma_{\mathrm{pzr}}=a_{L_{4}}\cdot L_{4}^{b_{9}}\\ \sigma_{\mathrm{pzr}}=a_{H_{4}}\cdot H_{4}^{b_{10}} \end{array}\right.$$
where *σ**_pzr_* is the stress at 50 μm from the edge of the cavity along the centerline, *a_i_* and *b_i_* are variable coefficients, respectively.

#### 3.2.2. Numerical Simulation

During the COMSOL simulation, we set the external load *P* to 200 MPa, the elastic modulus *E* to 169 GPa, and the Poisson’s ratio *ν* to 0.28. A series of stress values can be obtained by changing the size of each parameter within a certain range, and then curve fitting is performed by MATLAB. For example, if *L*_0_ and *H*_0_ are set in the range of (500, 1500) and (400, 800), respectively, the relationship between stress and size parameters can be obtained:(16)$$\begin{array}{l} L_{0}:\sigma_{pzr}=4.8\times 10^{5}\cdot L_{0}^{-1.1}\\ H_{0}:\sigma_{pzr}=231.8\cdot H_{0}^{0.0465} \end{array}$$

As shown in Figure 5d, it is found that only the relationship between the beam width *W*_3_ and the observed position stress does not conform to the form of a power function. In order to simplify the model, *W*_3_ is not introduced in the final relationship. Meanwhile, it can be found that the equivalent stress is the smallest when the beam width is between 40 and 60 μm. Because piezoresistance is arranged on the beam, in order to increase the stress near the piezoresistance, the beam width *W*_3_ should be less than 40 μm or greater than 60 μm.

By analogy, the approximate relationship between structural parameters and stress can be obtained:(17)$$\sigma_{pzr}=a\cdot \frac{L_{1}^{0.9926}\cdot H_{1}^{0.2406}\cdot H_{2}^{0.3316}\cdot H_{3}^{0.1503}\cdot H_{4}^{0.1094}\cdot H_{0}^{0.0465}}{L_{0}^{1.1}\cdot L_{2}^{0.3612}}$$

After the parameters of Table 2 are simulated by finite element, the coefficients can be obtained by putting them into Equation (17):(18)$$\sigma_{pzr}=1.426\times \frac{L_{1}^{0.9926}\cdot H_{1}^{0.2406}\cdot H_{2}^{0.3316}\cdot H_{3}^{0.1503}\cdot H_{4}^{0.1094}\cdot H_{0}^{0.0465}}{L_{0}^{1.1}\cdot L_{2}^{0.3612}}$$

#### 3.2.3. Parameter Optimization Design

By Equation (18) and Figure 5a,b, it can be found that the length and width of the upper cover *L*_0_ and the length and width of the upper cavity *L*_1_ have the greatest influence on the stress. The stress increases as the length and width of the upper cover and cavity increase (the length and width dimensions of the lower cavity and the upper cavity are the same). As the size of the upper cover and the base increase and the stress decreases. In other words, the increase in the outer lateral size of the core leads to the decrease in the internal stress. The thickness *H*_0_ of the upper cover has the least influence on the stress. Therefore, by first considering the sizing optimization of the upper cover and upper chamber, it can be found that the thinner the sidewall thickness (*L*_0_–*L*_1_) of the upper cover, the greater the equivalent force; however, too thin a sidewall will cause sidewall deformation and excessive stresses, even resulting in failure rupture. 

As shown in Figure 5, when *L*_0_ is more than 800 μm, its maximum effective force is stable at about 1 GPa, while when *L*_1_ is more than 700 μm, the maximum stress is too large, and finally exceeds the silicon fracture stress of 7 GPa. Therefore, taking into account the core overload capacity and maximum sensitivity, *L*_0_ for 900 μm, *L*_1_ for 500 μm.

The maximum stress of *L*_2_ is at 1200 μm, so *L*_2_ for 1200 μm (Figure 5c). Due to the limitation of *W*_3_ on the size and arrangement of the piezoresistive resistance, the design of *W*_3_ in this paper is 80 μm, and the principle of selecting other parameters is to maximize the stress in the allowable range of dimensions. Therefore, 800 μm for *H*_2_, 20 μm for *H*_3_, and 400 μm for *H*_4_ (Figure 5).

The piezoresistive resistors are P-type lightly doped in areas of high stress, and the piezoresistive resistors are placed on the four ends of the cross in the cross structure. Since the stress does not fluctuate much on the cross (Figure 6b), this design chooses to arrange piezoresistance at a distance of 100 μm from the edge. Each piezoresistance is designed for a value of 5 KΩ and the length of the piezoresistance is limited by the width dimension of the beam, which is 120 μm in length, 40 μm in each section, and 5 μm in width.

## 4. Results and Discussion

### 4.1. Simulation Results

The three-dimensional model of the cross-structured all-silicon sensor is established through COMSOL. The structural parameters are shown in Table 3, and the boundary conditions of the stress field and the current field are set (Table 4). The maximum applied pressure is 200 MPa and the supply voltage is 5 V. Finally, the relationship between the output voltage of the optimized structure sensor and the externally applied pressure is obtained (Figure 7).

According to the simulation results, the final sensitivity of the sensor can be calculated by Equations (10) and (11):(19)$$S=\frac{87.74-0}{(200-0)\times 5}=87.74\mathrm{uV}/V/\mathrm{MPa}$$

$NL_{i}$: By calculating the nonlinear error at each pressure *P*, where the largest value is the nonlinear error of the transducer. Finally, the maximum nonlinear error at 80 MPa was found to be 0.54%. That is, the nonlinear error is 0.54%.

### 4.2. Discussion

The CCSB structure proposed this article, All-Si Bulk [21], All-Si Bulk with Trench [22], Si-Glass Bulk [19], Si Diaphragm-type [11], Resistance strain gauge [3] and Fiber Bragg grating [4] were compared in performance, and the results are shown in Table 5.

At present, the sensors for rock stress monitoring include strain gauge pressure sensors [3] and optical fiber pressure sensors [4]. Compared with traditional strain gauge sensors, the CCSB structure MEMS pressure sensor has a larger range and higher sensitivity (silicon has better piezoresistive effect than metal materials), compared with FBG pressure sensor, CCSB structure MEMS pressure sensor has a larger range. For pressure sensors that are also based on MEMS technology, we have compared those given in the following subsections.

#### 4.2.1. Sensitivity and Non-Linear Error

The CCSB structure of this study has improved sensitivity and nonlinearity based on achieving high loads, and the All-Si Bulk-type is less affected by stresses due to thermal expansion than Si-Glass Bulk-type. As shown in Table 5, by comparing with the All-Si Bulk-type structure, the sensitivity of the CCSB structure is increased from 79 μV/V/MPa to 87.74 μV/V/MPa. Because the bottom cavity and cross structure are added to the All-Si Bulk structure, the stress at the bottom of the cavity is improved compared to the single cavity All-Si Bulk-type structure (piezoresistive area).

In addition to the advantages of silicon diaphragm-type structure in terms of sensitivity, the sensitivity of the CCSB structure is better than that of All-Si Bulk-type, all-silicon bulk-type with trench, and silicon-glass bulk-type. Comparing several structures, the key to improving the sensitivity of the sensor is to increase the stress in the piezoresistive region, and cavity or groove structures can allow for stress concentration, but complex movable structures can cause manufacturing difficulties and pose problems for the sensor’s resistance to overload and stability. Simulation results show that the sensor base contains an inner cavity structure, which improves the stress in the piezoresistive region, so the sensitivity is improved. However, the excessive stress leads to the decrease in the overload capacity of the sensor and the excessive strain in the cavity, which also leads to a decrease in the linearity and overload capacity.

#### 4.2.2. Structure and Pressure

Since the sensor is subjected to hydrostatic pressure all around except the bottom, the membrane at the bottom of the cavity strains, and with the cross structure, this strain can be reduced to improve the load resistance of the cavity and smooth out the stress changes on the membrane (Figure 8). The cross structure reduces the strain on the diaphragm in the piezoresistive region and improves the sensor’s shock and overload resistance.

Si diaphragm-type structures are commonly used in the low-pressure range, while in the high-pressure range CCSB structures can currently withstand pressures up to at least 800 MPa with good overload resistance, comparable to that of All-Si Bulk-type structures. The CCSB structure allows for a gentle distribution of stresses across the diaphragm, which facilitates piezoresistive placement and reduces the non-linear effects of piezoresistive position bias during the manufacturing process. With the addition of the cross structure, the large strains in the piezoresistive placement area are suppressed, relieving the stress concentration at the cavity edge, and the stress distribution on the piezoresistive surface is smoother and less nonlinear. The reduction in maximum stress results in increased load and overload capacity of the sensor. The CCSB structure pressure sensor features in the cross structure can improve the sensitivity of the sensor and allows the sensor to achieve a better overload capacity. In order to improve the bearing capacity, the key is the design of a cap, which can well avoid the diaphragm directly bearing large stress and strain, and improve the bearing capacity by indirectly measuring the stress.

## 5. Conclusions

In order to optimize sensitivity and linearity, this paper proposes a high-sensitivity MEMS piezoresistive pressure sensor with a CCSB structure. The COMSOL finite element simulation is used to predict the change of piezoresistive stress under different structural parameters, and the relationship equation between each structural parameter and stress is determined through the curve fitting method, which provides guidance for the design of the CCSB structure sensor. Through the finite element simulation, the CCSB and All-Si Bulk, grooved All-Si Bulk, Si-Glass Bulk and the Si Diaphragm structure were compared and analyzed. The CCSB structure not only improves the sensitivity and linearity of the sensor, but also has good anti-overload capability. Through finite element simulation, it is basically suitable for monitoring the internal stress of rock mass.

At present, the sensitivity of the sensor still has room for improvement. The next research work will focus on the selection of materials with higher piezoresistance coefficient, the optimal design of the shape of the piezoresistor, and the sensor packaging and experiments suitable for rock mass stress monitoring application scenarios. Because of the high cost of making sensors, this paper mainly optimizes the design of sensors from the perspective of design simulation, and verifies the design through simulation. In the future work, the use of semi-physical simulation to verify the experiment will be considered, and finally, the physical objects will be manufactured and verified.

## Figures and Tables

**Figure 1 sensors-22-07593-f001:**
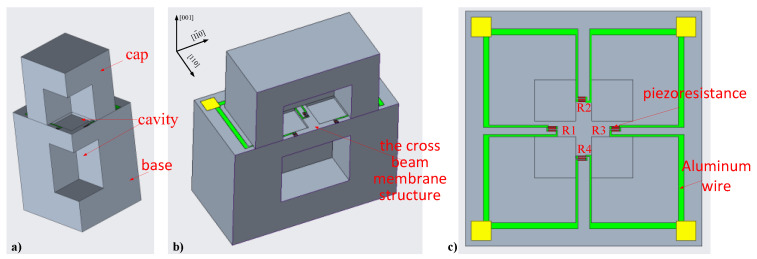
Schematic diagram of the optimized all-silicon pressure sensor structure, (**a**) 1/4 of FEA model, (**b**) 1/2 of FEA model, and (**c**) top view of the device layer.

**Figure 2 sensors-22-07593-f002:**
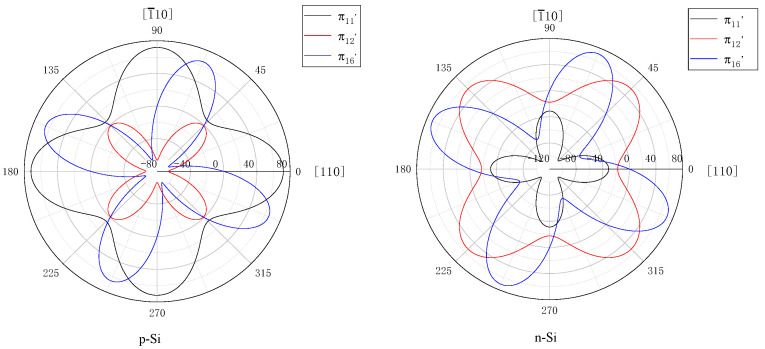
The p-type and n-type piezoresistance coefficients vary with direction on the (100) plane, and the radial coordinate is the piezoresistive coefficient (10^−11^/Pa). The circumferential scale is from 0 to 360 degrees. According to Equation (3), to maximize the relative resistivity, the p-type piezoresistance chooses the <110> crystal orientation arrangement, and the n-type piezoresistance chooses the <100> crystal orientation arrangement.

**Figure 3 sensors-22-07593-f003:**
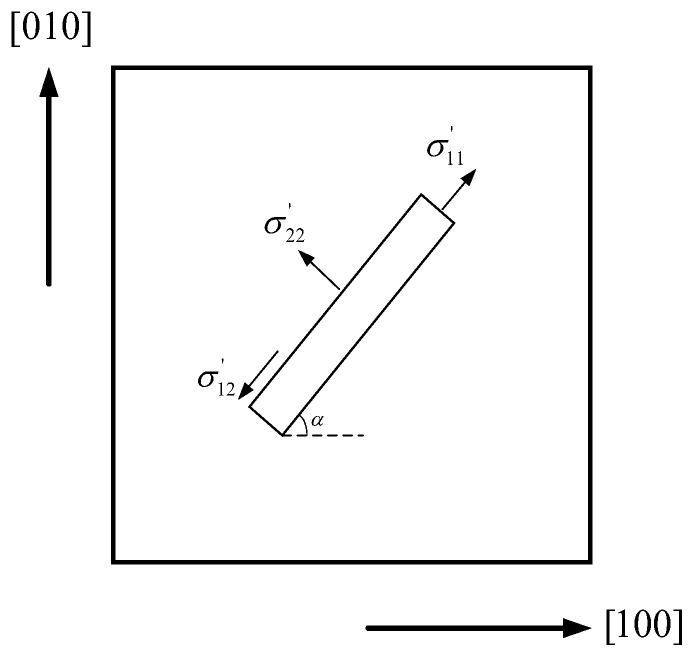
Diagram of piezoresistive forces on the plane (100).

**Figure 4 sensors-22-07593-f004:**
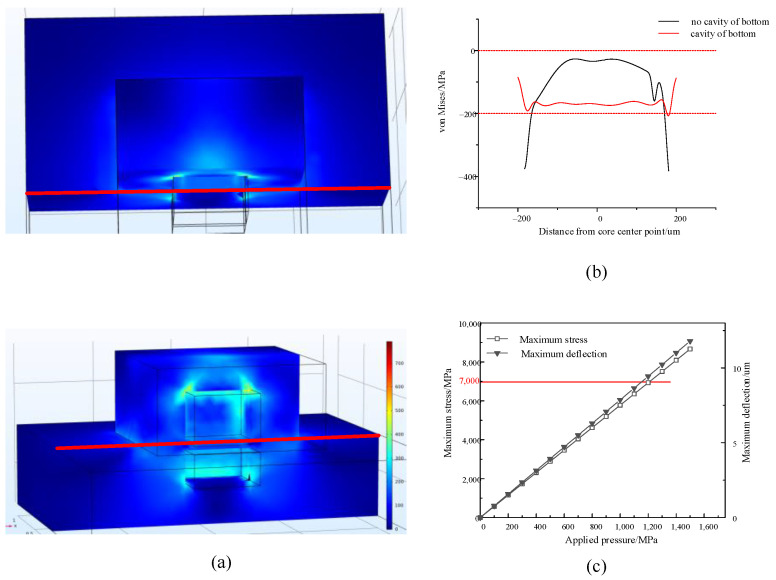
(**a**) Simulation model of the bottomless cavity and bottomed cavity structure. (**b**) The equivalent stress distribution along the centerline of the base surface (the piezoresistive doped area is in the upper cavity). Except for the cavity, the two models have the same structure size, the externally applied hydrostatic pressure is 200 MPa, and the elastic modulus *E* is 1.69 × 10^11^ Pa. (**c**) Relationship between the external load of the CCSB structure and the maximum stress and displacement of the sensor.

**Figure 5 sensors-22-07593-f005:**
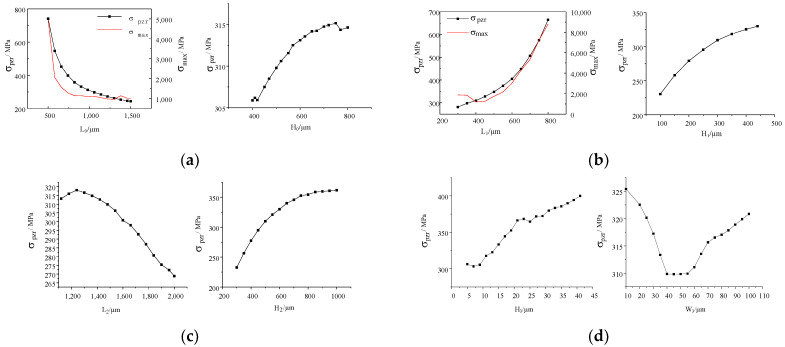
The relationship between structure size (**a**) *L*_0_, *H*_0_, (**b**) *L*_1_, *H*_1_, (**c**) *L*_2_, *H*_2_, (**d**) *H*_3_, *W*_3_, (**e**) *H*_4_ and equivalent stress.

**Figure 6 sensors-22-07593-f006:**
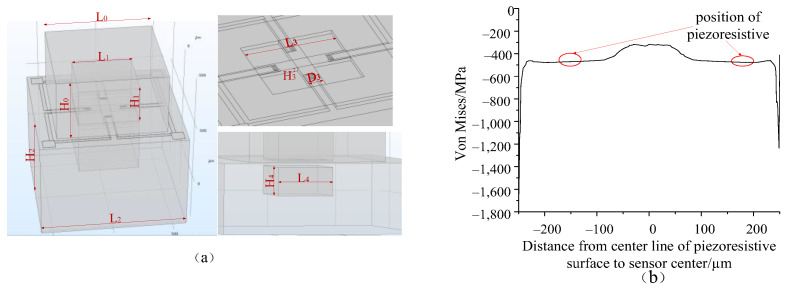
(**a**) Schematic diagram of the dimensions of the all-silicon bulk-type structure, and (**b**) stress curves in the piezoresistive placement area.

**Figure 7 sensors-22-07593-f007:**
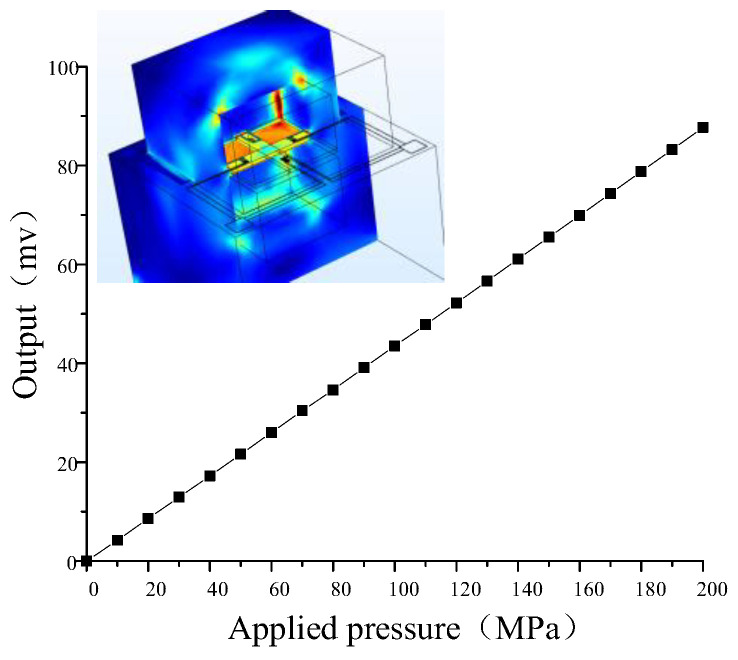
Sensor external load and output characteristic curve.

**Figure 8 sensors-22-07593-f008:**
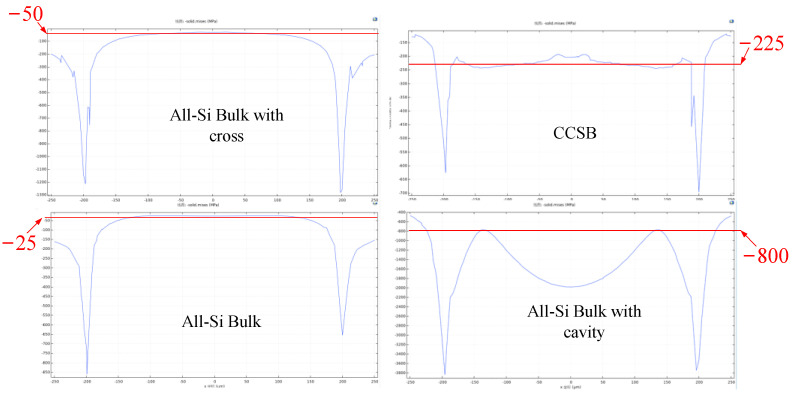
Chart of the effect of the stresses corresponding to the bottom cavity and cross structure.

**Table 1 sensors-22-07593-t001:** Piezoresistive coefficient of monocrystalline silicon (10^−11^/Pa).

	*π* _11_	*π* _12_	*π* _44_
p-Si	6.6	−1.1	138.1
n-Si	−102.2	53.4	−13.6

**Table 2 sensors-22-07593-t002:** The size of CCSB structure size (μm).

*L* _0_	1000	*L* _1_	400	*L* _2_	1500	*L* _3_	400	*L* _4_	400
*W* _0_	1000	*W* _1_	400	*W* _2_	1500	*W* _3_	40	*W* _4_	400
*H* _0_	500	*H* _1_	300	*H* _2_	500	*H* _3_	10	*H* _4_	200

**Table 3 sensors-22-07593-t003:** Optimized cross structure parameters.

Structure Parameters	Size/μm
*L* _0_	900
*H* _0_	500
*L*_1_(*L*_3_,*L*_4_)	500
*H* _1_	300
*L* _2_	1200
*H* _2_	800
*D* _3_	80
*H* _3_	20
*H* _4_	400

**Table 4 sensors-22-07593-t004:** Simulation parameters.

Parameters	Value
P	200 MPa
E	169 GPa
ν	0.27
VS	5 V

**Table 5 sensors-22-07593-t005:** Performance comparison of pressure sensors.

Structure	P (MPa)	Power Supply	Sensitivity (μV/V/MPa)	Non-Linear Error (%FSS)	Other Features	References	Application Scenarios
CCSB	200	5 V	87.74	0.54		this paper	Rock mass stress monitoring
All-Si Bulk	200	5 V	79			[21]	Oil exploration
All-Si Bulk with Trench	200	5 V	49			[22]	Oil exploration
Si-Glass Bulk	200	5 V	22.24	0.35		[19]	
Si Diaphragm-type	150	1.5 mA	1.1126 mV/MPa	0.3		[11]	Petrochemical industry
Resistance strain gauge	10 Kg		N/A	0.5	*^1^ K = 2	[3]	Geotechnical stress measurement
Fiber Bragg grating	50	N/A	N/A		0.3~0.5 × 10^−3^ mm (Accuracy)	[4]	Rock mass stress monitoring
Fiber Bragg grating	−3~13 KPa	N/A	N/A	1.0	2 Pa (Resolution)	*^2^ Luna OS9100	Rock mass stress monitoring

*^1^ K is the instrument sensitivity factor (nominal sensitivity factor), that is, the ratio of the relative change of the strain gauge resistance to its axial strain. *^2^ Luna OSA9100 is designed by Luna Corporation (https://lunainc.com/, accessed on 23 June 2021).

## Data Availability

Not applicable.

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
