# Peer review of "A Novel High-Sensitivity MEMS Pressure Sensor for Rock Mass Stress Sensing"

_sensors, 2022, doi:10.3390/s22197593_

Round 1

Reviewer 1 Report

In this work, the authors present high-sensitivity micro-electromechanical system (MEMS) piezoresistive pressure sensor that can be used for rock mass stress monitoring. They build analytical models and use FE simulations to optimize the structure parameter. I think the quality of this work is good and suggest a publication with minor revisions. Please use high-resolution pictures of the model in Comsol. Current pictures look like cropped images from Comsol rather than exported with high resolution. The scale bar is not clearly shown. 

Author Response

Thanks for your suggestions. The model in Figure 1 has been exported from the 3D modeling software with a high-resolution image, and Figure 6. has been exported by COMSOL with high resolution.

In addition, we have also actively revised some contents as follows:

  1. We have corrected the grammatical errors using the “Track Changes” function in the revised manuscript.
  2. We have checked the whole manuscript and corrected the misspelling.
  3. We have revised our manuscript according to the comments of other reviewers.
  4. We have updated the formula number and literature number.

We have tried our best to improve the manuscript and made some changes in the manuscript. These changes will not influence the content and framework of the paper. We appreciate your warm work earnestly, and hope that the correction will meet with approval.

We would like to express our great appreciation to you for comments on our manuscript.

Thank you and best regards.

Honghui

Reviewer 2 Report

This article proposes a novel, highly sensitive MEMS based piezoresistive pressure sensor with a cross, dual-cavity, and all-silicon bulk-type (CCSB) structure, which is claimed to be suitable for high-pressure range for rock mass stress monitoring.

Analytical as well as numerical modelling and simulations have been carried out using COMSOL and MATLAB.

Paper is well written with detailed literature review carried out by the authors on the material selection as well as structure design of pressure sensors.

1. The author has reportedly designed a sensor for high pressure application scenarios such as rock fracture stress and oil and gas exploitation. So, what are the generally accepted range of stress/pressure values for rock mass stress distribution evaluation?

2, Is there any link of natural frequency of the sensor with the sensitivity improvement? Can the stress bearing capacity be improved by increasing the stiffness of the sensor structure?

3. The author has incorporated a “cap” like structure in the design that increases the stress bearing and overloading capacity of the senor. Is “cap” like structure a novel one or is it employed from the literature? A more objective way is needed to highlight the contribution of author in this research.

4. Apart from the structural parameters variation, what are the key mechanical properties of the structure that have been improved for increasing the stress bearing capacity? What is the order of the system?

5. To evaluate the relations for change in sensitivity, author has incorporated generalised equations of piezo-resistive effect in section 2.2 which again puts question on the real contribution of author in this research. More in-depth analysis of piezo-resistive behaviour against the  is required. 

6. Author has employed bottom cavity structure in sensor design which reportedly increases the sensitivity of the sensor as compared to bottomless structure. However, no relation of sensitivity with stress has been described for both the structures.

7. Author has developed approximate theoretical relationships by using the combination of finite element calculation and curve fitting and has derived the expression of stress in equation 12.This again limits the author’s contribution in the theoretical formulation of this article.

8. Only geometrical parameters are varied and the consequent effect on the stress is evaluated. Author has not explored the mechanical aspects that are behind the stress bearing and overloading capacity improvement.

9. Author has claimed a sensitivity improvement by using the optimised design but has not presented any experimental validation of this work.

10. Proofreading is needed for this manuscript to improve the readability avoid the repeatability of sentences.

11. Picture quality needs to be improved for effective presentation. For example, Figure 1 does not entail the detailed views of the sensor structure hence more views must be inserted to improve for better illustration for the reviewer

The overall work needs a revision to improve the objectiveness in author’s contribution.

Reviewer 3 Report

The manuscript contains the research on high-sensitivity MEMS pressure sensor for rock mass 2 stress sensing. Sensor shows high sensitivity of 87.74 μV/V/MPA and a low 17 nonlinearity error of 0.28 % full-scale span (FSS) within the pressure range of 0-200 MPa. The manuscript can be accepted after moderate grammatical and English changes. 

Author Response

Thanks for your suggestions. The author has completely revised the English grammar and expression of the paper to meet the publication requirements.

In addition, we have also actively revised some contents as follows:

  1. We have corrected the grammatical errors using the “Track Changes” function in the revised manuscript.
  2. We have checked the whole manuscript and corrected the misspelling.
  3. We have revised our manuscript according to the comments of other reviewers.
  4. We have updated the formula number and literature number.

We have tried our best to improve the manuscript and made some changes in the manuscript. These changes will not influence the content and framework of the paper. We appreciate your warm work earnestly, and hope that the correction will meet with approval.

We would like to express our great appreciation to you for comments on our manuscript.

Thank you and best regards.

Honghui